# Forest Type and Site Conditions Influence the Diversity and Biomass of Edible Macrofungal Species in Ethiopia

**DOI:** 10.3390/jof8101023

**Published:** 2022-09-28

**Authors:** Gonfa Kewessa, Tatek Dejene, Demelash Alem, Motuma Tolera, Pablo Martín-Pinto

**Affiliations:** 1Sustainable Forest Management Research Institute, University of Valladolid, Avda. Madrid 44, 34071 Palencia, Spain; 2Department of Forestry, Ambo University, Ambo P.O. Box 19, Ethiopia; 3Wondo Genet College of Forestry and Natural Resources, Hawassa University, Shashemene P.O. Box 128, Ethiopia; 4Ethiopian Forestry Development, Addis Ababa P.O. Box 3243, Ethiopia; 5Ethiopian Forest Development Bahir Dar Center, Bahir Dar P.O. Box 12627, Ethiopia

**Keywords:** edaphic variables, edible mushrooms, natural forests, plantation forests, sporocarp yield

## Abstract

Ethiopian forests are rich in valuable types of non-wood forest products, including mushrooms. However, despite their nutritional, economic, and ecological importance, wild edible mushrooms have been given little attention and are rarely documented in Ethiopia. In this study, we assessed mushroom production levels in natural and plantation forests and the influence of climate and environmental variables on mushroom production. Sporocarps were sampled weekly from July to August 2019 at a set of permanent plots (100 m^2^) in both forest systems. We analyzed 63 plots to quantify sporocarp species’ richness and fresh weight as well as to elucidate the degree of influence of forest types and site conditions, including soil and climate. Morphological analyses were used to identify fungi. In total, we recorded 64 wild edible fungal species belonging to 31 genera and 21 families from the plots established in the natural and plantation forests. A significantly greater total number of edible fungi were collected from natural forests (n = 40 species) than from plantations. Saprotrophs (92.19%) were the dominant guild whereas ectomycorrhizal fungi represented only 6.25% of species. Ecologically and economically important fungal species such as *Agaricus campestroides*, *Tylopilus niger*, *Suillus luteus*, *Tricholoma portentosum*, and *Morchella americana* were collected. The sporocarp yield obtained from plantation forests (2097.57 kg ha^−1^ yr^–1^) was significantly greater than that obtained from natural forests (731.18 kg ha^−1^ yr^–1^). The fungal community composition based on sporocarp production was mainly correlated with the organic matter, available phosphorus, and total nitrogen content of the soil, and with the daily minimum temperature during collection. Accordingly, improving edible species’ richness and sporocarp production by maintaining ecosystem integrity represents a way of adding economic value to forests and maintaining biological diversity, while providing wood and non-wood forest products; we propose that this approach is imperative for managing Ethiopian forests.

## 1. Introduction

The Ethiopian highlands were once covered with dense natural high forests [1,2]. However, anthropogenic influences associated with the expansion of farming and human habitation have brought about deforestation and forest degradation [2], resulting in dramatic changes in the abundance and extent of native forest types. What remains is highly fragmented and frequently modified by non-native fast-growing trees such as *Eucalyptus* and *Pinus* species [3]. However, an important feature of these natural and plantation forests is their richness in valuable types of non-timber forest products (NTFPs) [4,5,6]. Various studies have revealed long lists of NWFPs, indicating their importance, contribution, and use by rural communities in Ethiopia [7,8]. If managed and conserved properly, NTFPs could potentially support the livelihoods of rural people by providing a source of income as well as food and medicine [7]. The most important NTFPs include honey, bees-wax, bamboo, coffee, spices, edible plant products such as fruits, seeds, and fodder, medicinal plants, various extractives and flavorings [7,9], and wild edible mushrooms [10,11].

More than 2000 fungi are known to produce edible sporocarps [12]. Edible wild mushrooms are not only a source of nutrition but also can be an important income generator [12,13], helping rural people to reduce their vulnerability to poverty and strengthen their livelihoods by providing a reliable source of revenue in many regions of the world [14,15,16]. In addition, edible mushrooms also form the basis of many manufactured products, including medicine, and are the focus of a new wave of tourism resulting from recreational programs linked to nature. Furthermore, wild mushrooms in general are important components of biodiversity in forest systems [17], playing an essential ecological role in forest communities. In particular, mycorrhizal associations have significant effects on nutrient and water uptake, growth, and plant survival [18,19], improving soil aeration and porosity [20] and resistance to pathogens [21], and also providing food for many organisms [22]. Saprotrophic fungi are essential for the decomposition of dead matter and, therefore, for nutrient cycling in forest ecosystems [23]. They are also important in the cycling of various elements, such as carbon, nitrogen, and oxygen [17].

There has been a tradition of ethnomycological usage among communities that dwell in Ethiopian forests [10,11,24]. However, despite all the benefits, wild mushrooms are the most neglected of Ethiopia’s NTFP resources. To date, there have been few studies of wild mushrooms and they are rarely documented [25,26], which may reflect that forest resource management has been primarily based on the production of wood products [8,9,27]. Wild mushroom formation is influenced by various factors, such as the forest type, host plant, nutritional status of the mycelium, and environmental factors [28,29,30]. Climatic variables also influence mushroom production because the development of fungal fruit bodies is dependent on soil temperature and the availability of surface water [31,32]. In addition, sporocarp composition is strongly determined by soil nutrients and chemical properties [33,34]. This is particularly the case for saprotrophic fungi, which are more dependent on their respective substrates than mycorrhizal fungi [35]. Forest management can, therefore, play a crucial part in shaping macrofungal communities because it can modify vegetation parameters such as tree density, canopy cover, understory plant communities, and soil conditions [30,33].

The negative impact of plantation forests is a common narrative in Ethiopia. Among these criticisms, the most cited is the lack of plant diversity in plantation forests [36,37]. However, our knowledge and understanding of fungal communities in plantation forests are limited compared to our knowledge of those in natural forests. Such information is essential to encourage forest management practices that include the conservation and production of valuable mushrooms through the adoption of a mycosilvicultural approach [38,39] for plantation and natural forest systems in Ethiopia. However, few studies have assessed the different environmental factors required for the production of wild edible mushrooms in these forest systems. To date, most have been performed at local scales and investigated a limited set of environmental variables. To gain a deeper understanding of the different environmental factors driving fungal communities as well as host plants, we should consider a wide range of variables from broad geographic areas. This should enable the development of strategies to manage different forests in different landscapes, to promote sporocarps production in the country. Thus, the main goal of this study was to assess the diversity of valuable macrofungal species in different areas of the country and compare the edible macrofungal species’ richness, sporocarp production, and community assemblages in native mixed forests and non-native fast-growing plantation forests. We hypothesized (1) that forest type is important for structuring macrofungal communities and the production of their sporocarps at small spatial scales in Ethiopian forest ecosystems. We expected that the macrofungal community would differ between the two forest types, resulting in an overall higher diversity value for study sites in natural forests because fungal diversity would be driven mainly by vegetation type, substrate availability, and other environmental variables [40,41,42]. However, the shorter rotation period of trees in plantation forests would also result in nutrient stress in these forests [43], which would favor only certain macrofungal species, such as ectomycorrhizal species [44]. Thus, we hypothesized that the overall sporocarp biomass produced in plantation forests would be higher than that in natural mixed forests, particularly those of potentially marketable species such as *Suillus luteus*. Differences in environmental variables such as edaphic and climate variables are the main driving forces for edible mushroom communities and, hence, would be important for the maintenance of their local production and diversity [45,46]. However, in the dry Afromontane region, water is scarce for a long period each year, implying that sporocarp development by some species would be limited. Thus, we also expected (2) that climate variables would govern fungal communities more than edaphic variables in our study areas [47]. When we set out, our specific objectives were: (i) to analyze the richness and production of sporocarps according to forest types, and (ii) to assess the influence of edaphoclimatic variables on taxa composition. The information generated from this study should highlight the economic value of edible mushrooms as NTFPs and encourage forest managers to manage these forests sustainably to promote sporocarp development and maintain the biological diversity of Ethiopian forests.

## 2. Materials and Methods

### 2.1. Study Sites

The study was conducted in five dry Afromontane areas in Ethiopia: the Taragedam, Alemsaga, Banja, Wondo Genet, and Menagesha Suba forests (Figure 1). Comprehensive descriptions of the forests are provided in Table 1. Some pictures of the forest types where the sporocarps collection was carried out in the study areas are provided in Figure 2 and Figure 3.

### 2.2. Experimental Design and Sporocarp Sampling

In total, 63 sample plots were established at seven sites, nine plots in each of the natural and plantation forests described in [53,54,55]. The plantation forests are located in the Wondo Genet and Menagesha Suba areas, while the natural forests are found in the Wondo Genet, Taragedam, Banja, and Alemsaga areas. The plantations mainly constitute the *Eucalyptus grandis*, *Pinus patula*, and *Pinus radiata* tree species (Table 1). The main tree species in the dry Afromontane natural forests include *Juniperus procera*, *Podocarpus falcatus*, *Hagenia abyssinica*, and *Olea africana* in the Wondo Genet natural forest area [27]. The dominant tree species in the natural forests of Taragedam, Banja, and Alemsaga, meanwhile, include *Maytenus obscura*, *Carissa edulis*, *Olea* sp., *Acacia abyssinica*, *Buddleja polystachya*, *Acacia nilotica*, *Albizia gummifera*, *Prunus africana*, and *Brucea antidysenterica.* These trees serve as the main sources of timber for the country [27] and thus indicate a need for the sustainable management of these forests. Each plot in each forest type was rectangular in shape (2 m × 50 m) and covered an area of 100 m^2^. Within each of the selected forest stands, we studied three different blocks with three plots per block. The plots were established at least 500 m apart and were laid out randomly in the forests to avoid confounding spatial effects inherent to such a plot-based design [56,57] and to reduce environmental heterogeneity [58]. The plots were analyzed as independent samples as suggested by Ruiz-Almenara [59]. All fungal fruit bodies found were harvested weekly during the major rainy season through July and August 2019. Fresh weight measurements were taken in situ using a digital sensitive balance (SF-400) to determine the fruit body production in kilograms per hectare per year. The number of sporocarps of each species in each plot was also recorded. Specimens were photographed in the field, and their morphological features and ecological characteristics were noted to facilitate taxonomic identification in the laboratory [60]. Specimens of each macrofungal species were taken to the laboratory and dried to preserve as herbaria specimens, then used for species identification purposes.

### 2.3. Species Identification and Characterization

In this study, morphological analyses were used for taxa identification. In the laboratory, fruit body tissues and spores were examined using an Optika B-350PL microscope (Optika, Pontenarica, Italy). Monographs were also used for taxa identification [61,62,63,64,65,66,67,68,69]. Up-to-date fungal taxa names and authors’ names were obtained from the Mycobank database (http://mycobank.org (accessed on 1 November 2020). Trophic levels were assigned to species using the recent classification compiled by Põlme et al. [70]. Fungal species’ edibility classification was accomplished by assessing the commercial importance of the collected species [71,72].

### 2.4. Environmental Data Collection and Analysis

To relate the composition of edible fungal taxa to edaphic variables, soil samples were collected from each of the sample plots established in each forest. After clearing and removing plant matter and debris from the soil surface, five soil cores were extracted from the center and the four corners of each plot using an auger (2 cm radius, 20 cm depth, and 250 cm^3^). The soil cores collected from each plot were pooled, and a composite, relatively homogeneous subsample of approximately 500 g from each plot was placed in a plastic bag for analysis. The soil pH was determined by analyzing a soil:water (1:2.5) suspension with the aid of a pH meter [73]. The organic carbon (C) content of the soil was determined using wet digestion [74]. The Kjeldahl procedure was used to determine the total nitrogen (N) content of the soil samples [75]. Sodium bicarbonate (0.5 M NaHCO3) was used as an extraction solution to determine the available phosphorus (P) [76]. The soil analysis was conducted by Water Works Design and Supervision Enterprises, a laboratory service sub-process, the soil fertility section at Addis Ababa, and the Amhara Water Works Design and Supervision Works Enterprise at Bahir Dar, Ethiopia.

In addition to the soil samples, the following climate variables were obtained for each forest from nearby meteorological stations: daily, mean, minimum, and maximum temperatures (°C), total annual precipitation (mm), and the average temperature (°C) and precipitation (mm) values for July and August of 2019 (i.e., the sporocarp collection season). 

### 2.5. Data Analysis 

Data were transformed when necessary to achieve the parametric criteria of normality and homoscedasticity. Macrofungal data were normalized by rarefying the abundance data to the smallest number of macrofungi per plot. In addition, data from environmental variables were scaled using base R and used for subsequent statistical analyses. The sporocarp biomass (kg ha^–1^ yr^–1^) was estimated for each forest. Differences in sporocarp production levels across forests were assessed using linear mixed-effects models (LME) [77], where a block (a set of plots at the same site in each forest) was defined as random and the forest was defined as a fixed factor. LME analyses were used to prevent false positive associations due to the relatedness structure in the sampling. Tukey’s test was later used to check for significant differences (*p* ≤ 0.05) between forests when needed.

Relationships between sporocarp composition and environmental parameters were visualized using non-metric multidimensional scaling (NMDS) based on an absence and presence species data matrix and environmental scaled data. A permutation-based nonparametric MANOVA (PerMANOVA) [78] using the Euclidean distance was performed to analyze differences in sporocarp communities between forest types and across the five sites. Isolines were also plotted on the NMDS ordinations for rainfall using the ordisurf function. Correlations between NMDS axes scores with explanatory variables were assessed using the envfit function in R. To assess the influence of edaphic, climate, and location variables on the fungal community, we performed a Mantel test (Bray–Curtis distance) on the total species matrix and scaled environmental parameters. 

## 3. Results

### 3.1. Edible Fungal Richness

In total, 64 edible fungal species belonging to 31 genera and 21 families were identified from the plots established in the natural and plantation forests. All these species belonged to the Basidiomycota, except for *Morchella* spp., which belonged to the Ascomycota. There was a significant difference in the total number of edible species found in each of the forest types (F = 7.23, *p* = 0.002). Forty edible species belonging to 22 genera and 15 families were found in the natural forests, while 16 species belonging to 16 genera and 12 families were found in plantation forests (Table 2). Eight edible species were common to both forest types. 

Saprotrophs were the dominant guild (92.19%; n = 59 species) followed by ectomycorrhizal fungi (6.25%; n = 4 species), with other guilds comprising 1.56% of the species (Table 2). Ecologically and economically important edible fungal species such as *Agaricus campestroides*, *Agaricus subedulis*, *Tylopilus niger*, *Suillus luteus*, *Tricholoma portentosum*, *Tricholoma saponaceum*, *Morchella americana*, and *Morchella anatolica* were collected (Table 2).

### 3.2. Sporocarp Production 

We found significant differences in the total edible sporocarp production between the two forest types (F = 4.293 *p* = 0.04; Figure 4A), with significantly greater mean sporocarp production levels in plantation forests (2097.57 kg ha^−1^ yr^–1^) than in natural forests (731.18 kg ha^−1^ yr^–1^). 

The average sporocarp production levels of edible species differed significantly among the five studied sites (F = 9.24; *p* = 0.0001), with the highest mean production levels recorded in the Menagesha Suba forest (7.49 kg ha^−1^). This value was significantly higher than that of the Wondo Genet (*p*_-MS_ _ *p*_-WG_ = 0.004), Banja (*p*_-MS_ _ *p*_-Ba_ = 0.0001), Alemsaga (*p*_-MS_ _ *p*_-Al_ = 0.000), and Taragedam forests (*p*_-MS_ _ *p*_-TG_ = 0.001). Mean sporocarp production in Wondo Genet forests (3.29 kg ha^−1^) was also significantly higher than that in Alemsaga forests (*p*_-WG_ _ *p*_-Al_ = 0.02) but was not significantly different from that recorded for the Taragedam (*p*_-WG_ _ *p*_-TA_ = 0.775) and Banja forests (*p*_-WG_ _ *p*_-TG_ = 0.2626). Mean sporocarp production levels in Alemsaga, Banja, and Taragedam forests did not differ significantly (*p* > 0.05; Figure 3B).

### 3.3. Sporocarp Composition and Environmental Variables

The perMANOVA analyses indicated that the two forest types differed significantly in their sporocarp composition (F = 5.343, R2 = 0.14, *p* = 0.001; Figure 5A). Explanatory variables categorized as edaphic, climate, and location parameters were correlated with fungal community composition (*p* < 0.05; Table 3). Of these, a Mantel test confirmed that location variables had a significantly stronger aggregate effect on the sporocarp composition of both forest types (*p* = 0.000) than climate (*p* = 0.0001) or edaphic variables (*p* = 0.0001). The significance of each explanatory variable and their aggregated contribution to differences in sporocarp composition are shown in Table 3.

When sites were analyzed separately, environmental variables such as N, P, OC, Tmin, and latitude were significantly correlated (*p* < 0.05) with the composition of edible sporocarps in the study areas. Of these variables, the Mantel test confirmed that location variables had a significantly stronger aggregate effect on sporocarp composition (r = 0.6419; *p* = 0.0001) than climate (r = 0.4378; *p* = 0.0001) and edaphic variables (r = 0.3166; *p* = 0.0001) when analyzed by site (Figure 4B). 

## 4. Discussion

In the most rural parts of Ethiopia, the local people are dependent on forest resources, either in the form of subsistence or as a cash income derived from NTFPs [7]. The collection of wild edible mushrooms by local people is a common practice, particularly in the southwestern parts of the country [11]. However, wild mushrooms are not considered important sources of food and medicine by rural communities in the northern part of the country [79]. This might be due to the continuing exodus of people from the countryside, which has meant that local communities are gradually losing their traditional knowledge, particularly about wild mushroom species. Furthermore, although a limited number of studies have reported the availability of wild mushrooms in Ethiopia and their importance as sources of food, medicine, and to some extent, income for local communities [10,11], information about the type of wild edible fungal species that are available, their potential production, and their status in different forest systems is scant [80]. This study is the first systematic survey focused on wild edible mushrooms, which was carried out in forests located in central and northern Ethiopia, where remnants of natural forests and plantations of exotic trees exist [81]. We collected a total of 64 edible fungal species from the study sites. The majority of the edible species (n = 40) were collected only from study sites in natural forests. 

We collected wild edible species that have both economic and ecological significance belonging to the genera *Morchella*, *Suillus*, and *Tylopilus* in plantation forests and *Tricholoma* in natural forests [82]. In addition, we found some *Agaricus* species and *Termitomyces*, which are known to be used by rural people in the southwest part of the country [26,83], and a *Schizophyllum* species, which is eaten by local people in southern Ethiopia [10,24]. Of these species, *Suillus luteus* is consumed by local people and is also sold in markets at a good commercial price in different developing countries, along with other NTFPs [44]. Furthermore, these kinds of mushrooms could help to sustain communities during periods of food scarcity, serving as an important source of nutrients for local people [82]. The use of wild mushrooms by rural people as a food source during lean periods has been documented in Ethiopian ethnomycological literature. In most cases, these species are collected for subsistence use [84]. However, in some places, mushrooms can provide households with additional income when sold in the markets. For example, in local markets in the southern and southwestern parts of the country, *Agaricus* sp. and *Termitomyces* sp. are available occasionally in association with other vegetal products, which the local people sell to earn some extra money to supplement their household income [83]. Therefore, the conservation and development of these kinds of valuable species deserve special attention given their possible role in increasing food security and income generation to subsidize rural household economies. In addition, as mushroom collection from wild habitats is seasonal, maintaining some of these edible species by means of local small-scale cultivation practices or in private forest areas would be very remunerative. Therefore, a strategy is needed for the adoption and cultivation of important species from the wild, which will not only increase their utilization but also create new sources of income for rural people and contribute to food security.

Although we collected more edible species from plots in natural forests than in plantation forests, sporocarp production levels were higher in plots in plantation forests than in natural forests. The greater number of species but lower sporocarp production levels in natural forests is unsurprising given that almost all the species found were saprotrophic and they were mainly composed of singleton taxa, with a small number of frequent species, which is in agreement with the findings of previous studies [85]. The saprophytic fungal species collected were also characterized by low levels of biomass production, in accordance with Gassibe et al. and Mediavilla et al. [54,86]. Nonetheless, they are relevant for decomposition processes and ecosystem functioning [87], particularly in tropical forest systems such as the natural forests in this study, where decomposition is rapid [88]. This may reflect the accumulation of favorable substrates, which is likely to enhance the richness [89] of these systems. The conspicuous sporocarps produced by these saprophytic fungi may also have favored the collection of this particular fungal class, although basidiomycete mycelia are reported to be everywhere in forests [90]. Overall, the sporocarp yield obtained from plantation forests (2097.57 kg ha^−1^yr^–1^) was significantly higher than that from natural forests (731.18 kg ha^−1^yr^–1^). Interestingly, almost 25% of the species collected from plantation forests were marketable species and have economic significance, including *Morchella* sp., *Suillus* sp., and *Tylopilus* sp. [12]. These species were characterized by their high levels of biomass production. For example, in Peru and Mexico, *Suillus* and *Morchella* species are commercial NTFPs produced in plantation forests. They guarantee the economic performance of those forests [44,91] and the livelihoods of local communities [92], thus providing incentives for farmers to plant and manage more plantations in their surroundings. Furthermore, in Mexico, *Morchella* species are also exported to generate income [44]. Although the overall biomass produced by sites in this study was low, the most productive site (Menagesha Suba) had mean production levels per stand of 7.49 kg ha^−1^, which suggests the potential production levels of edible sporocarp species in forests with similar conditions in this area. This also provides a starting point in terms of broadening the management of forests for the production of NTFPs such as edible mushrooms in Ethiopia, depending on the location and type of forest. 

Although natural forests produced lower sporocarp yields than plantation forests and had fewer marketable species, the overall yields and species were still valuably enhanced by plantations of exotic conifer species. A recent study [85] in the northern part of Ethiopia indicated that the overall land connectivity of natural forests with that of plantations provided important ectomycorrhizal species such as *Tricholoma* and *Suillus* in natural forest systems, indicating that such forest management activities could create important microniches with suitable resources and abiotic conditions to support more valuable mushroom species [93]. Thus, enrichment of natural forest systems through planting diverse tree species could potentially offer suitable habitats to enhance the richness and productivity of valuable edible species in natural forests in the study areas.

As we hypothesized, distinct fungal communities were observed in the two forest types. Fungal communities in natural forests were characterized by a large number of species, which may have been due to the greater spatial heterogeneity of the soil in natural forests compared with plantation forests. A heterogeneous soil environment and high rainfall levels create microhabitats in which saprotrophic species should be able to find the resources they require to survive in natural forests. The vegetation composition also impacts the composition of edible fungal species via the quantity and quality of the organic inputs, which mainly affect the saprotrophic community structure [94]. Most of the species found in the natural forests were associated with litter decomposition, which is typically characteristic of tropical forests [42]. However, some specific species such as *Termitomyces* and *Tricholoma* species were exclusively found in natural forests. The genus *Termitomyces* comprises a group of gilled mushrooms that have formed a termitophilic association with a particular family of termites, the Macrotermitinae (Isoptera), which are commonly found in Africa in places with a dry and humid climate [95]. This might be because our sampling sites in the natural forest areas were generally classified as dry Afromontane forest areas, characterized by high humidity and prolonged dry seasons [1]. These conditions might favor the occurrence and formation of distinct fungal community compositions in natural forest systems. Distinct fungal community compositions were also observed for plantation forests. Ectomycorrhizal fungi characterized the fungal composition of plantation forests along with some saprotrophic species. *Tylopilus* and *Suillus* species were site-exclusive species that were significantly more abundant in plantation forests comprised mainly *Pinus* than in other plantation forests. Some mushroom species, such as *Agaricus campestroides*, *Coprinellus domesticus*, *Leucoagaricus holosericeus*, *Hygrophoropsis aurantiaca*, and *Leucocoprinus cepistipes,* were common in both forest systems, indicating that these genera might be characterized as generalists. 

Studies have shown that different environmental variables govern the composition of fungal species and that different fungal taxa are likely to respond to edaphic variables in different ways, depending on their characteristics [96,97] and, in turn, the composition of fungal communities is directly correlated with soil parameters [98]. In this study, organic matter, P, and N were significantly correlated with the whole edible fungal species community dataset. This is likely to be because organic matter influences the fungal community through its impact on the water-holding capacity of the soil and on nutrient availability [99]. Thus, organic matter may favor more fungal assembly in an area, particularly saprotrophic fungi. Furthermore, the finding that N was an important factor correlated with fungal taxa compositions is in accordance with previous studies [28,35,100] that noted the influence of N on fungal distribution patterns. These studies reported that fungi showed community specialization toward more N-rich soil sites. This might be because nitrogen can influence the formation of extraradical mycelium in the soil and play a vital role in sporocarp formation [101]. Other studies have also noted that fungal communities adapt to more nitrogen-rich sites [100,102]. Furthermore, the microclimate directly influences ecological processes and reflects subtle changes in ecosystem functioning, particularly in forests where the majority of the identified edible species are saprotrophic and depend on a suitable microclimate for their growth and production. In that context, a mosaic forest management scheme is needed that considers both timber production and edible wild mushroom production. Such as scheme must uphold the environment variables needed to create suitable habitats, with variable microclimates to promote diverse sporocarps. This could increase the value of Ethiopia’s remnant natural forests and provide incentives for forest owners to sustainably manage and conserve the forests’ resources in different forms. 

## 5. Conclusions

Although our data were based on only one year of sampling during the rainy season in July and August 2019, we conclude that the richness of edible mushrooms is relatively higher in natural forests. The majority of these species are saprophytic and characterized by the production of small sporocarps. Although only a small number of edible species were recorded in plantation forests, sporocarp production levels in these forests were high. We also observed that some species specifically characterized each forest, and there was also a noticeable presence of valuable species in both forest types that could be potentially marketed in rural areas, providing forest managers and local people with supplementary incomes. Species such as *Termitomyces* and *Tricholoma* were exclusively found in natural forests, while *Suillus luteus* was found in plantation forests. However, environmental variables such as climate, spatial parameters, and edaphic parameters were found to affect the composition and, thus, sporocarp production in the study areas. In this regard, we found that the relative contributions of the spatial and climatic parameters were greater than those of the edaphic-specific variables, which significantly affected sporocarp production and species composition. However, a mosaic landscape, mixing natural and plantation forests, could provide timber and high levels of edible mushroom production. Accordingly, this could contribute to the conservation of remnant natural forests, which have high biodiversity values. Thus, a management scheme is needed that combines timber and mushroom production, which could provide economic and ecological benefits, especially in natural forest systems, which are characterized by excessive deforestation and forest degradation. Furthermore, the scheme should consider the influence of different environmental conditions, to create habitats that offer a suitable environment with variable microclimates for the promotion of diverse edible fungi and higher levels of sporocarp production, particularly in natural forests where the majority of the identified edible species were characterized by low levels of sporocarp production. 

## Figures and Tables

**Figure 1 jof-08-01023-f001:**
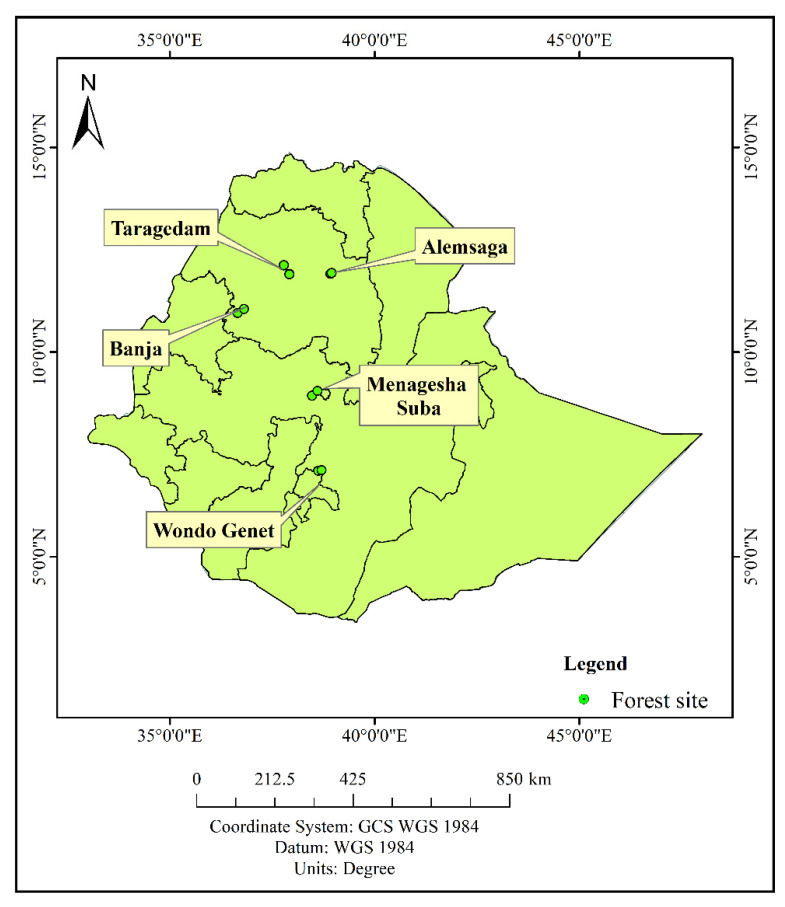
Map showing the locations of forests in which the study plots were located.

**Figure 2 jof-08-01023-f002:**
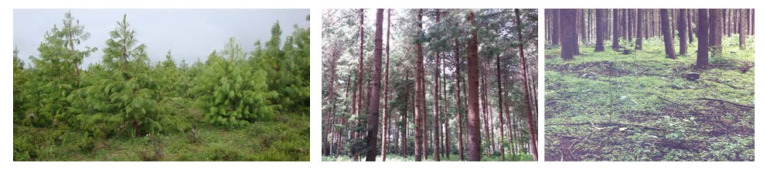
Plantation of *Pinus patula* where sporocarps collection was carried out in Wondo Genet (photo credit: Tatek Dejene).

**Figure 3 jof-08-01023-f003:**
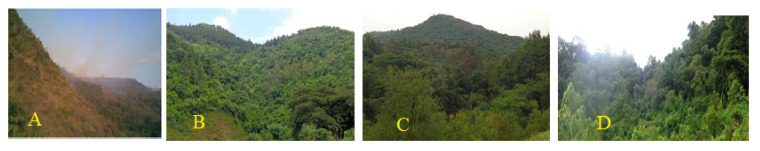
Dry Afromontane forests in (**A**) Wondo Genet, (**B**) Taragedam, (**C**) Alemsaga, and (**D**) Banja where sporocarps collections were conducted (photo credits: 3A, Tatek Dejene; 3B–3D, Demelash Alem).

**Figure 4 jof-08-01023-f004:**
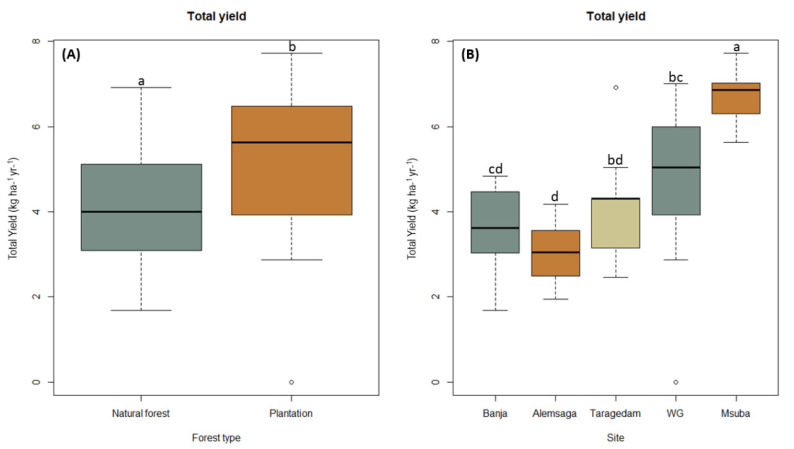
Production of sporocarps at five study sites in Ethiopia according to the forest type (**A**) and site (**B**). Data are presented as boxplots showing the maximum and minimum values. The bar in the box is the standard deviation of the mean. Values with the same letter are not significantly different. Msuba, Menagesha Suba; WG, Wondo Genet.

**Figure 5 jof-08-01023-f005:**
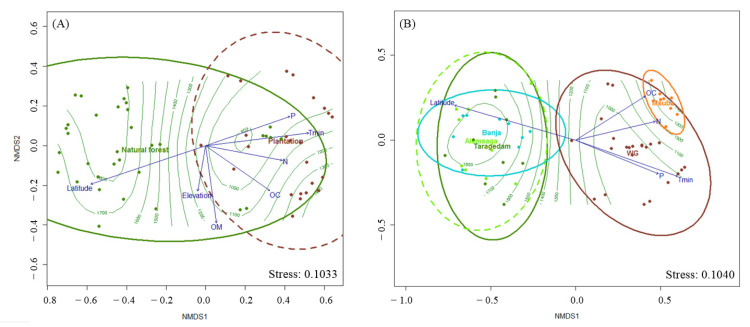
Non − metric multidimensional scaling (NMDS) ordination graph with fitted explanatory variables based on dissimilarities calculated using the Euclidean distance of sporocarp composition from plots in the two forest types (**A**) and by study sites (**B**), with rainfall displayed as isolines (fitted LOESS model R2 = 0.86). Arrows represent the environmental variables that were most significantly (*p* < 0.005) related to ordination. The five ellipses indicate the plots in the five study areas (Msuba, Menagesha Suba; WG, Wondo Genet). Explanatory variables are shown in blue: OM, organic matter; Tmin, minimum daily temperature; N, total nitrogen; P, available phosphorus; OC, organic carbon.

**Table 1 jof-08-01023-t001:** Comprehensive descriptions of the forests and the study sites.

Forest	Geographical Coordinates	Altitude (m asl)	MAP (mm)	MAT (°C)	Vegetation Types	Reference
Taragedam	12°06′59″–12°07′25″ N and 37°46′14″–37°47′02″ E	2062–2457	1300	20.4	Natural forests	[48]
Alemsaga	11°54′30″–11°56′00″ N and 37°55′00″–37°57′00″ E	2100–2470	1484	16.4	Natural forests	[49]
Banja	10°57′17″–11°03′05″ N and 36°39′09″–36°48′25″ E	1870–2570	1215.3	17.7	Natural forests	[50]
Wondo Genet	7°06′–7°07′ N and 38°37′–38°42′ E	1600–2580	1210	20	Natural forests and *Pinus patula* and *Eucalyptus grandis* plantation forests	[51]
Menagesha Suba	8°56′–9°03′ N and 38°28′–38°36′ E	2200–3385	1100	16	*Pinus radiata* plantation forests	[52]

Note: MAP, mean annual precipitation; MAT, mean annual temperature.

**Table 2 jof-08-01023-t002:** List of wild edible fungal species collected from study sites in natural and plantation forests, Ethiopia.

Species	Family	T	N	P
*Agaricus augustus* Fr.	Agaricaceae	S	x	
*Agaricus bitorquis* (Quél.) Sacc.	Agaricaceae	S	x	
*Agaricus campestris* L.	Agaricaceae	S	x	
*Agaricus campestroides* Heinem. and Gooss. -Font.	Agaricaceae	S	x	x
*Agaricus moelleri* Wasser	Agaricaceae	S	x	
*Agaricus murinaceus* Bull.	Agaricaceae	S	x	
*Agaricus subedulis* Heinem.	Agaricaceae	S	x	x
*Ampulloclitocybe clavipes* (Pers.) Redhead, Lutzoni, Moncalvo and Vilgalys	Hygrophoraceae	S	x	
*Armillaria heimii* Pegler	Physalacriaceae	P	x	
*Auricularia auricula*-*judae* (Bull.) Quél.	Auriculariaceae	S	x	
*Calvatia cyathiformis* (Bosc) Morgan	Lycoperdaceae	S	x	
*Calvatia gigantea* (Batsch) Lloyd	Lycoperdaceae	S	x	
*Calvatia subtomentosa* Dissing and M. Lange	Lycoperdaceae	S		x
*Clitocybe carolinensis* H.E. Bigelow and Hesler	Tricholomataceae	S	x	
*Clitocybe cistophila* Bon and Contu	Tricholomataceae	S	x	
*Clitocybe foetens* Melot	Tricholomataceae	S	x	
*Clitocybe fragrans* (With.) P. Kumm.	Tricholomataceae	S	x	
*Clitocybe geotropa* (Bull. ex DC.) Quél.	Tricholomataceae	S	x	
*Coprinellus domesticus* (Bolton) Vilgalys, Hopple and Jacq. Johnson	Psathyrellaceae	S	x	x
*Coprinopsis nivea* (Pers.) Redhead, Vilgalys and Moncalvo	Psathyrellaceae	S	x	
*Coprinus comatus* (O.F. Müll.) Pers.	Agaricaceae	S	x	
*Coprinus lagopus* (Fr.) Fr.	Psathyrellaceae	S	x	
*Coprinus micaceus* (Bull.) Fr.	Psathyrellaceae	S	x	
*Coprinus niveus* (Pers.) Fr.	Psathyrellaceae	S	x	
*Craterellus ignicolor* (R.H. Petersen) Dahlman, Danell and Spatafora	Hydnaceae	S	x	
*Crepidotus applanatus* (Pers.) P. Kumm.	Crepidotaceae	S	x	
*Crepidotus mollis* (Schaeff.) Staude	Crepidotaceae	S	x	
*Gymnopilus pampeanus* (Speg.) Singer	Strophariaceae	S		x
*Hygrocybe chlorophana* (Fr.) Wünsche	Hygrophoraceae	S	x	
*Hygrophoropsis aurantiaca* (Wulfen) Maire	Hygrophoropsidaceae	S	x	x
*Laetiporus sulphureus* (Bull.) Murrill	Laetiporaceae	S	x	
*Lentinellus cochleatus* (Pers.) P. Karst.	Auriscalpiaceae	S	x	
*Lepista sordida* (Schumach.) Singer	Tricholomataceae	S		x
*Lepista sordida* var. lilacea (Quél.) Bon	Tricholomataceae	S		x
*Leucoagaricus americanus* (Peck) Vellinga	Agaricaceae	S	x	
*Leucoagaricus holosericeus* (J.J. Planer) M.M. Moser	Agaricaceae	S	x	x
*Leucoagaricus leucothites* (Vittad.) Wasser	Agaricaceae	S	x	x
*Leucoagaricus purpureolilacinus* Huijsman	Agaricaceae	S	x	
*Leucoagaricus rubrotinctus* (Peck) Singer	Agaricaceae	S	x	x
*Leucocoprinus birnbaumii* (Corda) Singer	Agaricaceae	S	x	
*Leucocoprinus cepistipes* (Sowerby) Pat.	Agaricaceae	S	x	x
*Lycoperdon perlatum* Pers.	Lycoperdaceae	S		x
*Lycoperdon umbrinum* Pers.	Lycoperdaceae	S		x
*Macrolepiota africana* (R. Heim) Heinem.	Agaricaceae	S		x
*Macrolepiota procera* (Scop.) Singer	Agaricaceae	S	x	
*Morchella cf americana* Clowez and C. Matherly	Morchellaceae	S		x
*Morchella anatolica* Isiloglu, Spooner, Alli and Solak	Morchellaceae	S		x
*Omphalotus illudens* (Schwein.) Bresinsky and Besl	Omphalotaceae	S		x
*Pholiota aurivella* (Batsch) P. Kumm.	Strophariaceae	S	x	
*Pleurotus luteoalbus* Beeli	Pleurotaceae	S	x	
*Pleurotus populinus* O. Hilber and O.K. Mill.	Pleurotaceae	S	x	
*Pleurotus pulmonarius* (Fr.) Quél.	Pleurotaceae	S	x	
*Polyporus badius* (Pers.) Schwein.	Polyporaceae	S		x
*Polyporus tenuiculus* (P. Beauv.) Fr.	Polyporaceae	S		x
*Polyporus tuberaster* (Jacq. ex Pers.) Fr.	Polyporaceae	S		x
*Schizophyllum commune* Fr.	Schizophyllaceae	S		x
*Suillus luteus* (L.) Roussel	Suillaceae	EM		x
*Termitomyces clypeatus* R. Heim	Lyophyllaceae	S	x	
*Termitomyces microcarpus* (Berk. and Broome) R. Heim	Lyophyllaceae	S	x	
*Termitomyces robustus* (Beeli) R. Heim	Lyophyllaceae	S	x	
*Termitomyces schimperi* (Pat.) R. Heim	Lyophyllaceae	S	x	
*Tricholoma portentosum* (Fr.) Quél.	Tricholomataceae	EM	x	
*Tricholoma saponaceum* (Fr.) P. Kumm.	Tricholomataceae	EM	x	
*Tylopilus niger* (Heinem. and Gooss. -Font.) Wolfe	Boletaceae	EM		x

Note: T, trophic groups; S, saprotrophic; P, parasitic; EM, ectomycorrhizal; N, natural forest; P, plantation forest.

**Table 3 jof-08-01023-t003:** Significance of explanatory variables for sporocarp composition in the two forest types. The numbers in bold indicate a highly significant effect of environmental variables (*p* < 0.001).

Sources	Contribution	Variables	Pseudo-F	*p*-Values
Edaphic variables	32.33%	OM	0.2824	0.001
		N	0.3057	0.001
		P	0.4184	0.001
		OC	0.3122	0.001
Climate	60.76%	Elevation	0.0762	0.088
		Tmin	0.5217	0.001
Spatial factors	64.19%	Latitude	0.7777	0.001

Note: OM, organic matter; N, total nitrogen; P, available phosphorus; OC, organic carbon; Tmin, minimum daily temperature.

## Data Availability

Not applicable.

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
