# Peer review of "Forest Type and Site Conditions Influence the Diversity and Biomass of Edible Macrofungal Species in Ethiopia"

_jof, 2022, doi:10.3390/jof8101023_

Round 1
Reviewer 1 Report
The authors compared the species diversity and biomass of edible macro-fungi in two type forests, which will provide foundation for managing Ethiopian forests. A few suggestions or comments were added in the pdf. Please check it.

Author Response
REF.: Manuscript ID: jof-1910512
Forest type and site conditions influence the diversity and biomass of edible macrofungal species in Ethiopia
Dear Editor,
According with the advice received from the editor to resubmit with a revision based on the reviewers’ comments, we enclosed the revised manuscript “Forest type and site conditions influence the diversity and biomass of edible macrofungal species in Ethiopia” for consideration. It has been revised, incorporating the corrections suggested by the reviewers and the editor. Changes are highlighted using Track Changes System, as suggested by the editor, in the revised manuscript. A response to the reviewer’s comments is enclosed too.
With kind regards,
- Gonfa Kewessa
- Tatek Dejene
- Demelash Alem
- Motuma Tolera
- Pablo Martín-Pinto
Dpto. de Producción Vegetal y Recursos Forestales
Escuela Técnica Superior de Ingenierías Agrarias de Palencia
Universidad de Valladolid
Edificio departamental E
Avda. Madrid 44, CP.- 34007
PALENCIA – SPAIN
Responses to the Reviewers
#Reviewer 1
Too many explanations of Introduction better make shorter
- Although the suggestion is interesting and well received. We think the context of the area and the study is no frequently studied and that is the reason to provide a complete and deep introduction, which is needed for a better comprehension of the impact of this research. We prefer not to reduce it. However, if it is considered as a major request, we could be of course open to do it in a further review.
Please provide some short information forest vegetation such as diversity of woody plants including forest types. I think that Taragedam, Alemsaga, Banja, Wondo Genet, and Menagesha Suba forests are name of place
- Dear Reviewer, thank you the positive suggestion here. Of course, the Alemsaga, Banja, Wondo Genet, and Menagesha Suba forests are name of place. However, in the text we have tried to explained in which area was the plantations and the natural forests in the methodology part of 2.2. In this study, the plantation forests were located in Wondo Genet and Menagesha Suba areas, while the natural dry afromontane forests are found in the Wondo Genet, Taragedam, Banja, and Alemsaga areas. The plantations are mainly constituted by Eucalyptus grandis (Wondo Genet), Pinus patula (Wondo Genet), and Pinus radiata (Menagesha Suba) tree species (Table 1 of the Ms). The main tree species in the dry Afromontane natural forests of Wondo Genet included Juniperus procera, Podocarpus falcatus, Hagenia abyssinica, and Olea africana. Whereas, the dominant tree species in the natural forests of Taragedam, Banja and Alemsaga included Maytenus obscura, Carissa edulis, Olea, Acacia abyssinica, Buddleja polystachya, Acacia nilotica, Albizia gummifera, Prunus africana, and Brucea antidysenterica. This is now included in the revised version of the manuscript.
If possible, please provide some photos of forests it is good for reader to know the forest type
- Included as suggested in figure 3
Please provide which brand and which country and city product
- Provided as suggested
If you get ITS sequences why you didn't provide NCBI number of the fungal species and would be nice if you provide NBCI of your species and put it to 2 table otherwise better to delete this molecular
- Dear reviewer, thank you for the suggestion here. We have removed this part in the methodology section.
Please add one subparagraph information taxonomic diversity of the fungi such how many families, orders and genera and which widespread and so on. It is better for reader to know that which edible fungal species are common there
- Included as suggested
Summary: We appreciate the positive comments you have towards our study. We hope that we have provided the necessary responses for your concerns to reconsider our manuscript towards an eventual acceptance for publication. Of course, we remain open to clarify any further concern that you might have.
Author Response
REF.: Manuscript ID: jof-1910512
Forest type and site conditions influence the diversity and biomass of edible macrofungal species in Ethiopia
Dear Editor,
According with the advice received from the editor to resubmit with a revision based on the reviewers’ comments, we enclosed the revised manuscript “Forest type and site conditions influence the diversity and biomass of edible macrofungal species in Ethiopia” for consideration. It has been revised, incorporating the corrections suggested by the reviewers and the editor. Changes are highlighted using Track Changes System, as suggested by the editor, in the revised manuscript. A response to the reviewer’s comments is enclosed too.
With kind regards,
- Gonfa Kewessa
- Tatek Dejene
- Demelash Alem
- Motuma Tolera
- Pablo Martín-Pinto
Dpto. de Producción Vegetal y Recursos Forestales
Escuela Técnica Superior de Ingenierías Agrarias de Palencia
Universidad de Valladolid
Edificio departamental E
Avda. Madrid 44, CP.- 34007
PALENCIA – SPAIN
*************
# Reviewer 2
I suggest the authors add the species diversity in two type forests, eg. how many species and genera in natural forests and plantation forests respectively.
- Dear reviewer, we appreciate the positive comments you have towards our work. We have provided the requested information in the result part of the revised manuscript.
Did you get ITS sequences of every species? or some species were identified by only morphology? Please make it clear?
- Dear reviewer, thank you for the suggestion here. We removed the molecular information in the methodology part. We tried to complement the morphological identification of some few species by sequencing but due to the lack information in these ecosystems this work was not successful. To avoid confusion, we have now removed this methodology as any result was used
Belonging to xxx genera, xxx families.
- Included as suggested.
Summary: We hope that we have provided the necessary responses for your concern to reconsider our manuscript towards an eventual acceptance for publication. Of course, we remain open to clarify any further concern that you might have.